# Identifying prognostic subgroups of luminal-A breast cancer using deep autoencoders and gene expressions

**Seunghyun Wang, Doheon Lee** [ID] *

Department of Bio and Brain Engineering, KAIST, Daejeon, Republic of Korea

* dhlee@kaist.ac.kr

**Data Availability Statement:** There are no primary data in the paper; all data are available on the original publications of databases (https://pubmed.ncbi.nlm.nih.gov/22522925/, https://pubmed.ncbi.nlm.nih.gov/23000897/), cBioPortal (https://www.

## Abstract

Luminal-A breast cancer is the most frequently occurring subtype which is characterized by high expression levels of hormone receptors. However, some luminal-A breast cancer patients suffer from intrinsic and/or acquired resistance to endocrine therapies which are considered as first-line treatments for luminal-A breast cancer. This heterogeneity within luminal-A breast cancer has required a more precise stratification method. Hence, our study aims to identify prognostic subgroups of luminal-A breast cancer. In this study, we discovered two prognostic subgroups of luminal-A breast cancer (BPS-LumA and WPS-LumA) using deep autoencoders and gene expressions. The deep autoencoders were trained using gene expression profiles of 679 luminal-A breast cancer samples in the METABRIC dataset. Then, latent features of each samples generated from the deep autoencoders were used for K-Means clustering to divide the samples into two subgroups, and Kaplan-Meier survival analysis was performed to compare prognosis (recurrence-free survival) between them. As a result, the prognosis between the two subgroups were significantly different (p-value = 5.82E-05; log-rank test). This prognostic difference between two subgroups was validated using gene expression profiles of 415 luminal-A breast cancer samples in the TCGA BRCA dataset (p-value = 0.004; log-rank test). Notably, the latent features were superior to the gene expression profiles and traditional dimensionality reduction method in terms of discovering the prognostic subgroups. Lastly, we discovered that ribosome-related biological functions could be potentially associated with the prognostic difference between them using differentially expressed genes and co-expression network analysis. Our stratification method can be contributed to understanding a complexity of luminal-A breast cancer and providing a personalized medicine.

## Author summary

Luminal-A breast cancer is the most frequently occurring breast cancer subtype. However, it shows high variability in prognosis, and more precise stratification is needed. In this paper, we identified two prognostic subgroups of luminal-A breast cancer, BPS-LumA and WPS-LumA. To this end, we used deep autoencoders which automatically generate

cbioportal.org/), and GTEx (https://gtexportal.org/
home). We have archived our code on GitHub at
https://github.com/BISLshwang/ISLA.

**Funding:** SW and DL are supported by the Ministry
of Science and ICT through the National Research
Foundation (NRF-2022M3A9B6017511). The
funders had no role in study design, data collection
and analysis, decision to publish, or preparation of
the manuscript.

**Competing interests:** The authors declare that they
have no competing interests.

informative latent features that represent essential properties of gene expressions. We
found that the two subgroups clustered using the latent features are significantly different
in prognosis. This prognostic difference was validated with the external luminal-A breast
cancer cohort. We showed that only latent features are able to discover the prognostic sub-
groups compared to gene expression profiles. In addition, we compare our results with
the two previous luminal-A breast cancer stratification method which are complementary
to each other. Finally, we suggested biological functions associated with the differentially
expressed genes between the two subgroups as potential molecular mechanisms which
results in the differences in the prognosis. We expect that our method could be used for
the personalized medicine of luminal-A breast cancer.

## Introduction

Personalized medicine is the ultimate goal of modern medicine [1]. The personalized medicine
arises from that millions of people are taking medications that will not help them. It is reported
that the top ten highest-grossing drugs in the United states help only between 1 in 25 and 1 in
4 of the people who take them [2]. This is resulted from a limitation of large-scale clinical trials
which cannot consider the individual characteristics of each patient [3]. Fortunately, due to
the recent development of high-throughput technology, there are attempts to infer the individ-
ual characteristics of each patient from various omics data and to reflect them in the treatment
and management of the diseases. In this context, breast cancer is one of the remarkable dis-
eases that personalized medicine has been realized in the clinic.

Breast cancer which is one of the leading causes of death among females [4], and it has been
discovered that breast cancer is a heterogeneous disease of which subtypes have different
molecular mechanisms and require different therapeutic strategies [5]. Traditionally, immuno-
histochemical markers (e.g., estrogen receptor (ER), progesterone receptor (PR), and HER2)
are used to stratify the breast cancer patients [6]. Recently, PAM50 is the most popular subtyp-
ing method which classifies breast cancer into several intrinsic subtypes (e.g., luminal-A, lumi-
nal-B, HER2-enriched, and basal-like) based on expression levels of 50 genes [7]. Moreover, it
is well-known that there is significant concordance between the immunohistochemistry-based
stratification and PAM50 [7].

Especially, luminal-A breast cancer is the most frequently occurring subtype which
accounts for about 60~70% of whole breast cancer, and it is characterized by hormone recep-
tor-positive (ER, PR) and HER2 receptor-negative [8]. Hence, endocrine therapies have been
considered as first-line treatments for luminal-A breast cancer. For example, aromatase inhibi-
tors (e.g., anastrozole, letrozole, and exemestane) interrupt estrogen production by inhibiting
the aromatization of androgens to estrogens [9]. On the other hand, SERM (selective estrogen
receptor modulator), such as tamoxifen, block the binding of estrogen and estrogen receptor
and SERD (selective estrogen receptor degraders), such as elacestrant and fulvestrant, inhibit
translocating estrogen receptor to the nucleus and degrade them [9]. However, even in the
luminal-A breast cancer subtype, some patients show intrinsic and/or acquired resistance to
these endocrine therapy [10], and the prognosis of luminal-A breast cancer is more variable in
comparison with the other breast cancer subtypes [11].

Consequently, this heterogeneity within luminal-A breast cancer has required a more pre-
cise stratification method which makes it possible to predict the prognosis and provide the per-
sonalized diagnosis and treatment. Recently, several previous studies suggested prognostic
subgroups of luminal-A breast cancer through machine learning and gene expressions. For

example, Netanely et al. clustered luminal-A breast cancer samples into two prognostic subgroups (LumA-R1 and LumA-R2) using the most variable genes in their expressions [12]. Later, Poudel et al. classified the luminal-A breast cancer into five subgroups based on expression levels of several marker genes which represent five different cell types (Enterocytes, Inflammatory, Stem-like, Goblet-like, and TA) [13], and they revealed that four out of five subgroups were significantly different in prognosis [14].

Even though the previous studies successfully identified the prognostic subgroups of luminal-A breast cancer, they have a limitation that they required human engineering to select the features (genes) for stratifications. For example, Netanely et al. took the top 2,000 genes which show the highest variability in the expressions, and Poudel et al. selected the marker genes according to significances of microarray analysis [15].

In the perspective of feature engineering, a deep learning which takes advantages of data-driven automatic feature learning can be a promising solution [16]. Especially, an autoencoder is an artificial neural network which aims for data dimension reduction and feature extraction [17]. The training purpose of the autoencoder is to generate latent features in the hidden layers, which can reconstruct the input features in the output layer. In consequence, the autoencoder automatically extracts and compresses essential properties of the input features, and generates the informative latent features in the hidden layers. Recently, Tan et al. trained the autoencoder using gene expressions of breast cancer, and they showed that the latent features were able to discriminate intrinsic subtypes of breast cancer [18]. Dwivedi et al. showed that disease modules could be discovered through the autoencoder trained with large gene expression datasets [19].

In this study, we identified two prognostic subgroups of luminal-A breast cancer. To this end, we trained deep autoencoders using gene expression profiles of luminal-A breast cancer to generate informative latent features of each sample and we discovered the subgroups through the latent features and unsupervised learning. Additionally, we showed that our method has important biological contributions in realizing precision medicine of luminal-A breast cancer. First, we demonstrated that our method is feasible in an independent test set, which is the most important part to translate the deep learning approaches to clinical practices. Furthermore, we proved that the latent features are more useful than gene expression profiles and features generated by traditional dimensionality reduction method (i.e., PCA) in terms of discovering the prognostic subgroups. In addition, we suggested potential molecular mechanisms which determine the prognostic difference using differentially expressed genes between the subgroups and weighted gene co-expression network analysis. Lastly, we compared our stratification with two previous luminal-A breast cancer stratification methods.

## Results

### The latent features generated from the deep autoencoders successfully identify the prognostic subgroups of luminal-A breast cancer

First of all, as we aimed to obtain informative latent features to identify prognostic subgroups of luminal-A breast cancer, we trained deep autoencoders which are able to automatically extract and compress important properties of gene expression profiles without additional human engineering. To this end, we obtained gene expression profiles of 679 and 415 luminal-A breast cancer samples from METABRIC and TCGA dataset to use them as training set and validation set, respectively. Before training the deep autoencoders, we chose the top 5,000 genes which show the highest variability across the samples based on the median absolute deviation (MAD) in the METABRIC dataset (S1 Table). Then, we renormalized the gene expression profiles in both datasets using min-max scaling and used them as input features of the

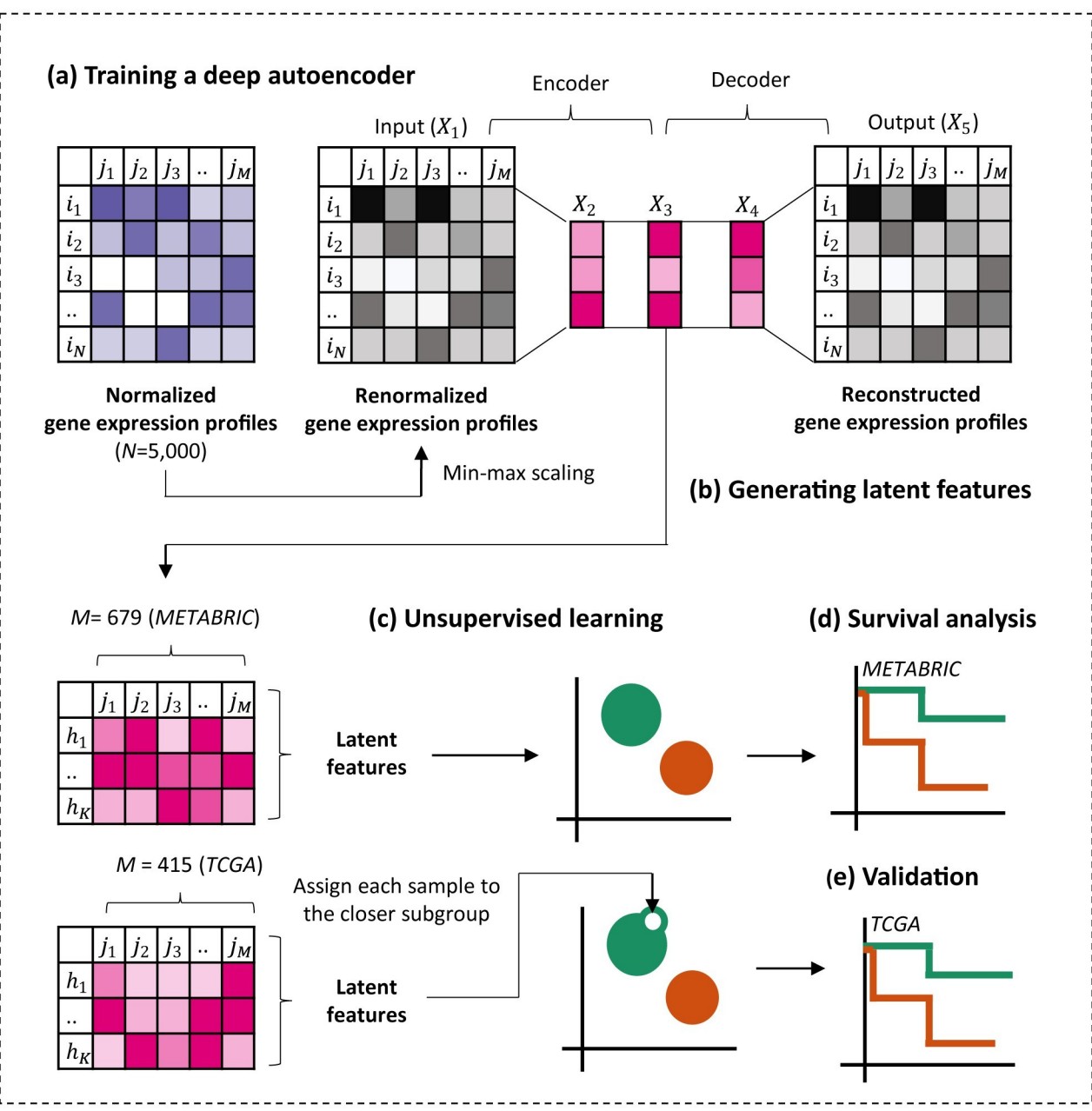

**Fig 1. Overall pipelines.** (A) $X_1$ are the renormalized gene expression profiles and $X_5$ is the reconstructed gene expression profiles from the latent features. N is the number of genes, and M is the number of samples. The deep autoencoders were trained using the renormalized gene expression profiles of luminal-A breast cancer in the METABRIC and the samples of the TCGA BRCA were used as the validation set. (B) The latent features of each 679 METABRIC sample were generated in the second hidden layer of deep autoencoders, and (C) the samples were divided into the subgroups using the latent features as input features of unsupervised learning. (D) The Kaplan-Meier analysis was performed to compare the prognostic differences (recurrence-free survival rate) between the subgroups, and (E) the prognostic differences were validated using the recurrence-free survival data of 415 TCGA samples.

deep autoencoders (Fig 1A and Methods). We trained the eight deep autoencoders with the different hidden layer size (16, 32, 64, 128, 256, 512, 1024, and 2048). We set the size of all three hidden layer size same to see whether the size of hidden layer affects the performance of deep autoencoders [19]. The performances of deep autoencoder were evaluated by mean

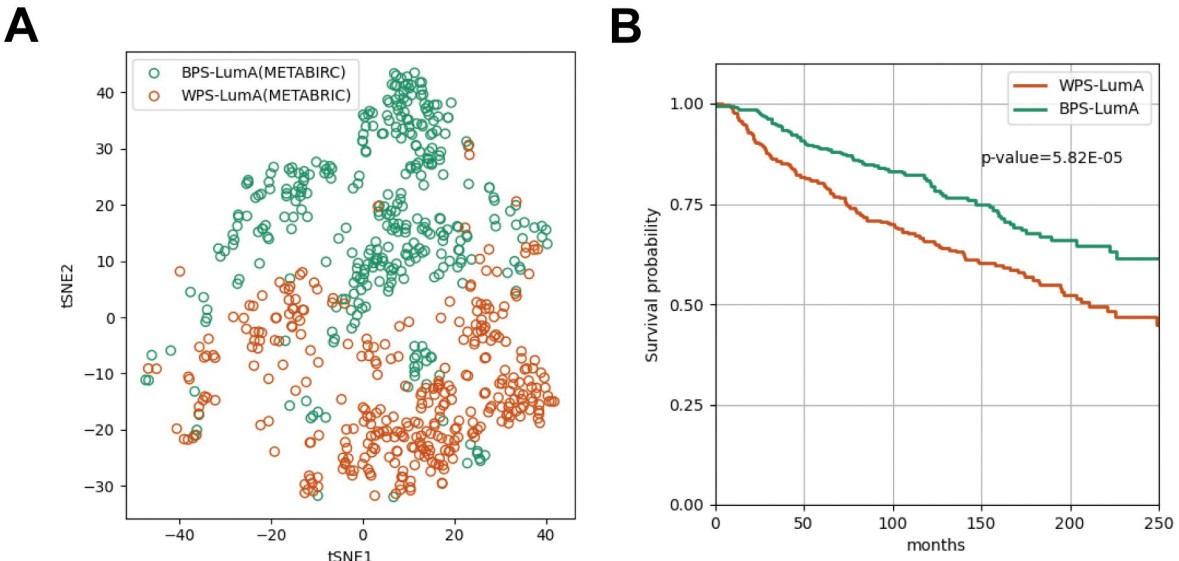

**Fig 2. The t-SNE (t-distributed Stochastic Neighbor Embedding) plot of 64-dimensional latent features and the Kaplan-Meier survival curve of 679 METABRIC luminal-A breast cancer samples.** (A) The t-SNE plot (dimension size = 2) of latent features generated from the deep autoencoders of 679 METABRIC luminal-A breast cancer samples. The samples assigned to the BPS-LumA (the better prognostic subgroup) and WPS-LumA (the worse prognostic subgroup) colored as green and orange, respectively. (B) The green and orange curve indicates the BPS-LumA and the WPS-LumA, respectively. The x-axis refers recurrence-free survival months and the y-axis refers survival probability.

squared error (MSE) between the renormalized gene expression profiles of the input layer and the reconstructed gene expression profiles of the output layer.

As we expected, the MSEs decreased continuously as the size of hidden layers increased in the training set (S2 Table), but the differences between the deep autoencoders were not that significant (MSE = 0.012±0.008). In the validation set, we observed that the differences between the deep autoencoders were much smaller than the training set (MSE = 0.075±0.003). More importantly, the MSEs decreased until the size of hidden layers increased to 128. However, they started to increase again from when the size of hidden layer is larger than 128. It indicates that the models might be overfitted to the training set. Therefore, we decided to use only the latent features obtained from the models of which the size of hidden layers are 16, 32, 64, and 128 in the following survival analysis.

The ultimate goal of our study is to identify distinct prognostic subgroups of luminal-A breast cancer. Hence, we clustered 679 METABRIC samples using the latent features generated from the deep autoencoders and unsupervised learning. We used K-means clustering for unsupervised learning. Then, we performed the Kaplan-Meier survival analysis to compare prognosis between the subgroups and the significance was evaluated by log-rank test.

As a result, the prognostic differences between the subgroups were stably significant (p-value < 0.01) when the samples were divided into two subgroups regardless of the dimensional size of latent features (S1 Fig), but the difference was the most significant when the dimensional size is 64 (p-value = 5.82E-05, Fig 2). However, when the samples were clustered into more than two subgroups using the 64-dimensional latent, and we found that some subgroup pairs did not show significant prognostic differences in the pairwise log-rank test. For example, when the samples were clustered into the three subgroups we observed that the samples belonged to the third subgroup (cluster3) did not show the significant prognostic differences with the samples belonged to the other subgroups (cluster1 and cluster2) (S3 Table and

S2 Fig). We observed similar tendencies when we divided the samples into four and five subgroups (S3 Table and S2 Fig).

Given these results, we concluded that the prognostic differences were the most distinct when 679 METABRIC samples were divided into the two subgroups using the 64-dimensional latent features (Fig 2A). In the Kaplan-Meier survival curve (Fig 2B), the first subgroup (n = 336) and the second subgroup (n = 343) show worse and better prognosis (S4 Table). In the following sections of this study, we named the better prognostic subgroup s 'BPS-LumA', and the worse prognostic subgroup as 'WPS-LumA'.

## The prognostic difference between BPS-LumA and WPS-LumA was validated in the independent dataset

To validate the prognostic differences between the BPS-LumA and the WPS-LumA, we applied our stratification method to an independent dataset. We used the 415 TCGA luminal-A breast cancer samples for validation. We generated the 64-dimensional latent features of each TCGA sample using the deep autoencoder trained with METABRIC dataset in the previous section, and each sample was assigned to the closer subgroup based on the distance between the sample and the centroid of the BPS-LumA and the WPS-LumA in the latent space (Fig 3A). Among 415 samples, 191 and 224 samples belonged to the BPS-LumA and the WPS-LumA, respectively (S5 Table). Interestingly, the samples belonging to the BPS-LumA showed significantly better prognosis than the other samples belonging to the WPS-LumA (p-value = 0.004; Fig 3B).

Interestingly, even though the two datasets used the different expression profiling platforms (The METABRIC and TCGA dataset use microarray and RNA-seq as expression profiling platforms, respectively), we proved that our method is applicable in the both datasets. To further explore these results, we measured the Spearman's rank correlation coefficient (SRCC) between the mean expression levels of individual genes in the both datasets and we observed that they are highly correlated (SRCC = 0.771, Fig 3C). Similarly, we measured the SRCC of the absolute median deviations of individual gene in the both datasets and we confirmed that they are also significantly correlated (SRCC = 0.595, Fig 3D). In addition to these correlations between two expression profiling platforms, the additional preprocessing to reduce the batch effects (e.g. min-max scaling) makes our methods successfully predicts the prognosis of the samples regardless of the expression profiling platforms.

## Only the latent features of deep autoencoders successfully identified the prognostic subgroups

Next, we wanted to compare our method with gene expression profiles and traditional dimensionality reduction method to show the usefulness of the deep autoencoders in terms of generating the informative latent features for the prognostic subgroup identification. Hence, similarly to when using the latent features, we divided the 679 METABRIC samples into the two subgroups using the expression profiles of whole 17,202 genes and the top 5,000 genes with the highest variability as the input features of K-Means clustering, respectively. Interestingly, we found that the prognostic difference between the subgroups identified using the whole 17,202 genes (p-value = 0.566; Fig 4A) and the top 5,000 most variable genes (p-value = 0.183; Fig 4B) were not significant.

In addition, we compared our method with traditional dimensionality reduction method. To this end, we projected the whole 17,202 genes and the top 5,000 most variable genes into 64- (the same dimensional size with the latent features that we used) and 2- (the most informative principal components) dimensional space using PCA (Principal Component Analysis),

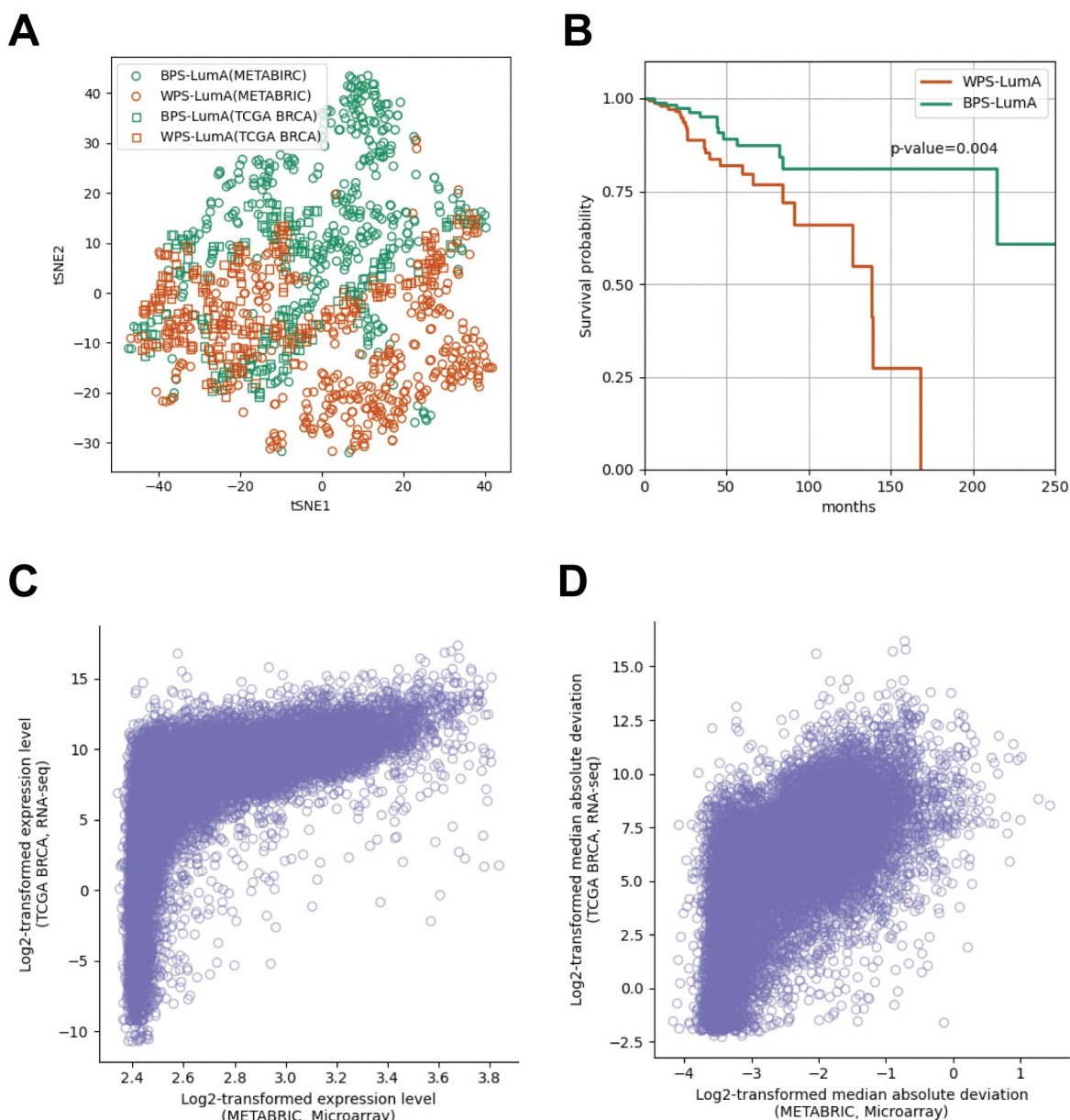

**Fig 3. The t-SNE plot of 64-dimensional latent features of all samples in the METABRIC and TCGA BRCA dataset and the Kaplan-Meier survival curve of and 415 TCGA BRCA luminal-A breast cancer samples.** (A) The t-SNE plot of all samples in the METABRIC and TCGA BRCA datasets. The samples of METABRIC and TCGA BRCA are denoted as circle and square, respectively. The samples assigned to the BPS-LumA and WPS-LumA colored as green and orange, respectively. (B) The Kaplan-Meier survival curve of the 415 TCGA samples which were assigned to the closer prognostic subgroups (BPS-LumA and WPS-LumA) in the latent space. (C) The mean log2-transforemd expression levels and (D) the log-transformed median absolute deviation of individual genes (N = 17,202) in the METABRIC (microarray, x-axis) and the TCGA (RNA-seq, y-axis) dataset were plotted using scatter plots. Each dot indicates individual genes in the (C) and (D).

respectively. As a result, we observed that none of them successfully discovered the distinct prognostic subgroups (Fig 4D–4F).

From these results, we confirmed that only the latent features were able to identify the prognostic subgroups p-value = 5.82E-05; Fig 2A). It indicates that the deep autoencoders more effectively extract the important properties from the complex gene expression profiles, which

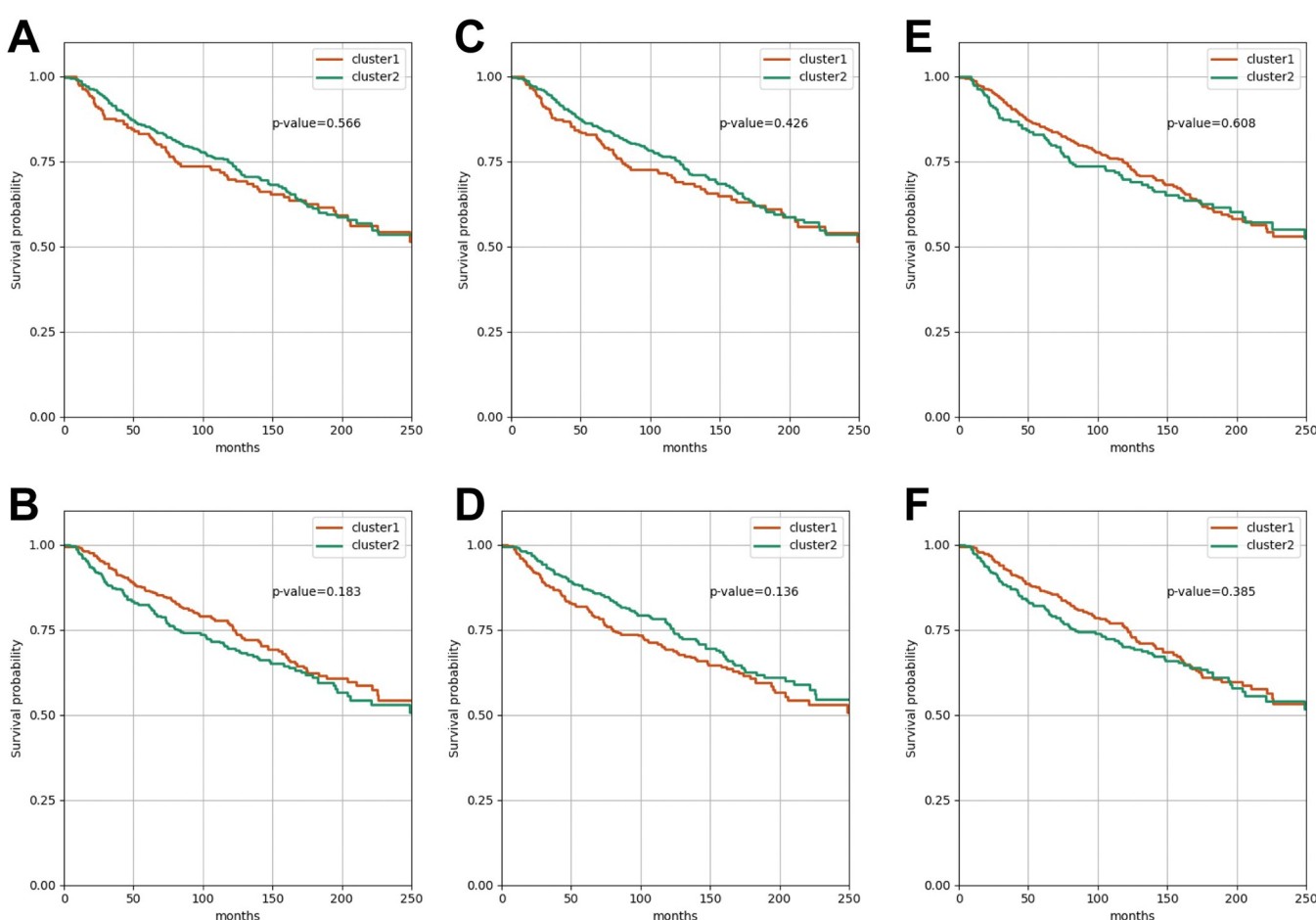

**Fig 4. The Kaplan-Meier survival curves of when the samples are clustered using the gene expression profiles and the low-dimensional features generated using PCA.** The Kaplan-Meier survival curve when the 679 METABRIC samples were divided into the two clusters using (A) the whole 17,202 genes, (B) the top 5,000 most variable genes, the 64-dimensional features (PCA) of (C) the whole 17,202 genes and (D) the top 5,000 most variable genes, and the 2-dimesional features (PCA) of (E) the whole 17,202 genes and (F) the top 5,000 most variable genes.

determine the prognosis of luminal-A breast cancer, than the traditional dimensionality reduction method and it helps to discover the distinct prognostic subgroups for precision medicine.

## Ribosome-related biological functions could be potentially associated with the prognostic difference between BPS-LumA and WPS-LumA

Next, we tried to figure out which biological functions potentially makes the prognostic differences between the BPS-LumA and WPS-LumA. To this end, among top 5,000 genes with the highest variability, we found the 548 differentially expressed genes (DEGs) between BSP-LumA and WPS-LumA in the both METABRIC and TCGA datasets through limma [20] (adjusted p-value<0.01, S6 and S7 Tables). Then, we constructed weighted co-expression network of breast tissue through WGCNA [21, 22], which is consisted of 23 co-expressed modules (Fig 5A and S8 and S9 Tables), using the 459 gene expression profiles of non-diseased breast tissue obtained from GTEx. We used co-expression network to analyze the DEGs because the malfunction of individual genes does not result in the dysfunction of biological systems due to robustness of the biological systems and the impact of the DEGs have to be analyzed at the system-level, not at the single-gene level [23–25]. Lastly, we measured proportion (%) of the genes overlapping with the DEGs in each co-expressed module.

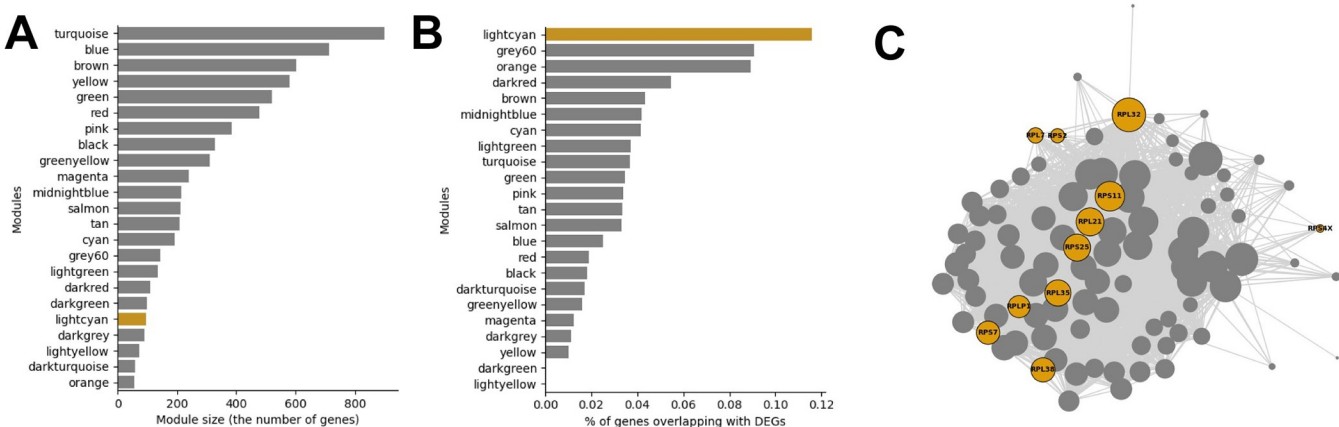

**Fig 5. The module size and the proportion of genes overlapping with the DEGs in each co-expressed module.** (A) The bar graph of the module size (the number of genes in each module). (B) The bar graph representing the % of genes overlapping with DEG. The lightcyan module is highlighted as yellow. (C) The network plot of the lightcyan module. The DEGs are colored as yellow and the node size indicates the connectivity of each node.

As a result, we observed that "lightcyan" co-expressed module includes the significantly large number of DEGs (11.6%, Fig 5B and S10 Table). We explored the biological functions related to the lightcyan module through the gene-set enrichment analysis [26] and we observed that the 95 genes included in the lightcyan module was associated with ribosome-related terms such as "rRNA metabolic process (GO:0016072)", "Ribosome (MAP 03010)", and "Cytoplasmic Ribosomal Proteins (WP477)" (S11 Table). Among the 95 genes in the lightcyan module, 11 genes were DEGs between BPS-LumA and WPS-LumA (Fig 5C).

We found that there are many literature evidences that report the associations between the ribosomal proteins (RPs) and cancer [27]. For example, the RP-MDM2-p53 signaling pathway is the most well-studied pathway which defines a role of ribosomal proteins in tumor suppressor gene p53 activation [28]. Recently, it was revealed that deregulation of some ribosomal proteins can promote breast cancer metastasis [29] and ribosome biogenesis could be potential therapeutic target to combat tamoxifen resistance [30]. Based on these results, the terms significantly associated with the DEGs and their activities in each subgroup could be considered as potential biological mechanisms that promote the prognostic differences between the BPS-LumA and WPS-lumA.

## BPS-LumA and WPS-LumA are complementary to the previous stratification methods of luminal-A breast cancer

Additionally, we wanted to see how much our method coincides with previous luminal-A breast cancer stratification methods: Netanely's method [12] and Poudel's method [14]. The Netanely's method suggested the two prognostic subgroup: LumA-R1 (poor prognosis) and LumA-R2 (good prognosis). The Poudel's method proposed the five heterocellular subgroups: stem-like (good prognosis), inflammatory (good prognosis), goblet-like (good/intermediate prognosis), TA (poor prognosis), and enterocyte. As a result, we confirmed that our method, which used the automatically generated latent features from the deep autoencoders, showed a tendency to be consistent with the two previous studies.

For example, our method was significantly concordant with the Netanely's method (p-value = 6.70E-05; Fig 6A and S12 Table). The LumA-R1 was composed of the more WPS-LumA samples (64.3%). On the other hand, the proportion of BPS-LumA was larger than the WPS-LumA in the LumA-R2 (55.9%). Similarly, our method was also concordant

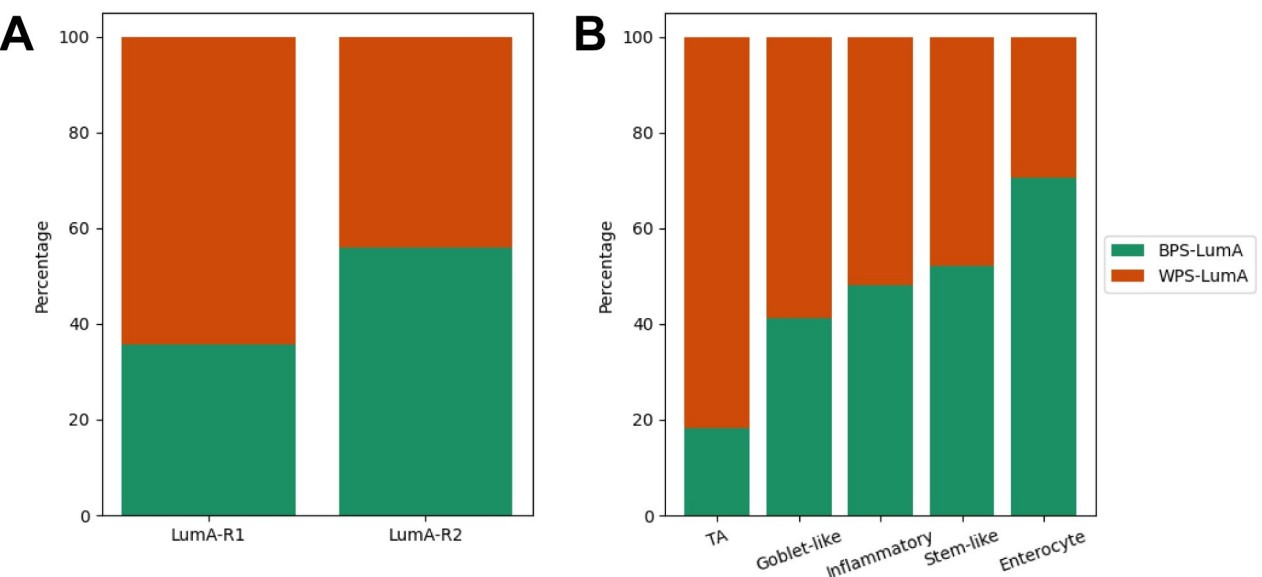

**Fig 6. The comparison with previous luminal-A breast cancer stratification methods.** (A) Netanely's method, (B) Poudel's method. The x-axis refers the subgroups identified in the previous studies and the y-axis indicates the percentage of the BPS-LumA and the WPS-LumA belonging to them. The BPS-LumA and WPS-LumA were colored as green and orange, respectively.

with the Poudel's method (p-value = 0.004; Fig 6B and S12 Table). We observed that the proportion of BPS-LumA samples decreased according to the prognosis of four subgroups in the Poudel's method: stem-like, inflammatory, goblet-like and TA subgroups. The 52.22% of stem-like subgroup belonged to the BPS-LumA. However, the proportion decreased to 48%, 41.18% and 18.18% in the inflammatory, goblet-like and TA subgroups, respectively. In addition, the Poudel's method did not provide the prognosis information of enterocyte subgroup, but interestingly it was composed of large proportion of BPS-LumA samples (70.68%).

## Discussion

In this study, we identified the two prognostic subgroups, BPS-LumA and WPS-LumA, of luminal-A breast cancer using the latent features generated from the deep autoencoders trained with the gene expressions. We validated the prognostic differences between the two subgroups using the independent dataset. We also showed that the latent features generated from the deep autoencoders are more useful to identify the prognostic subgroups than the gene expression profiles and the features generated from the traditional dimensionality reduction method (i.e. PCA).

Consequently, the remaining challenge is to develop more effective therapeutic strategies for each subgroup, especially for the patients who belongs to the worse prognostic subgroup. The stratification method can further maximize its power when there is an appropriate treatment for each subgroup. For example, patients with hormone receptor-positive breast cancer (e.g. luminal-A and luminal-B) receive an endocrine therapy as a first-line treatment. On the other hand, dual HER2 blockade (e.g. Trastuzumab and Pertuzumab) is recommended to hormone receptor-negative and human epidermal growth factor 2-positive breast cancer (e.g. HER2-enriched) [8]. Like this, to develop the appropriate therapeutic strategies for each subgroup identified in our study is our top priority to solve in the near future, and the enriched pathways which we found using the differentially expressed genes between the two subgroups could be a starting point. We observed that ribosome-related terms were significantly related with the differentially expressed genes between BPS-LumA and WPS-LumA and literature

evidences indicate that proteins involved in these biological functions could be considered as potential drug targets [28–32] for the patients who belong to the worse prognostic subgroup and are resistant to the endocrine therapy.

In addition to the DEG analysis, there are recently lots of efforts to develop explainable and interpretable deep leaning models to overcome "black-box" problem [33, 34], especially for biomedical and healthcare domain [35, 36]. These efforts could be helpful to interpret the biological meanings of the latent features generated in our method. Alternatively, even though network-based or pathway-based methods have some limitations that our knowledge about human interactome is still incomplete [37, 38] and the interaction information derived from experimental assays (e.g. yeast two-hybrid) and computational inference usually do not fully consider the context-specificity [39, 40], they could suggest more solid evidences to interpret deep learning approaches when the prior knowledges are combined with the deep learning models (e.g., visible neural network [41]). Likewise, the approaches to increase the interpretability of deep learning models could accelerate the development of novel therapeutic options for the patients of poor prognosis.

Lastly, we showed that our and the previous methods are significantly coincident and they could complement each other for a more precise stratification through further study. For example, in the Poudel's method, the samples belonged to goblet-like subgroups are regarded as intermediate prognosis, which are ambiguous than the other subgroups. Interestingly, we discovered that 41.18% and 58.82% of them are belonged to the BPS-LumA and WPS-LumA in our method. This result indicates that even though the samples show similar goblet cell-like signatures, they could show different characteristics in other biological pathways, such as ribosome-related pathways that we discovered with the DEG analysis. In this perspective, comprehensively considering the results of our and the previous methods can be helpful to more precisely define the disease states of the patients.

To sum up, we successfully developed the precise stratification method of luminal-A breast cancer which is able to predict the prognosis. Given that luminal-A breast cancer is the most frequently occurring breast cancer and the prognosis varies from patient to patient due to the endocrine resistance, our method could be helpful to stratify the patients and prepare alternative treatment options according to the predicted prognosis.

## Materials and methods

### Collecting gene expression profiles and recurrence-free survival data of METABRIC and TCGA BRCA

Two breast cancer datasets, METABRIC (the Molecular Taxonomy of Breast Cancer International Consortium) [42, 43] and TCGA (The Cancer Genome Atlas) BRCA [44], were collected. We downloaded all gene expressions, recurrence-free survival data and PAM50 subtype data of the both datasets from cBio cancer genomics portal except the PAM50 subtype data of the TCGA dataset [45]. We obtained the PAM50 subtype data of the TCGA dataset from the supplemental information of original publication [44]. In the case of the gene expression profiles, we downloaded normalized gene expression profiles (median Z-scores) of the both datasets instead of raw gene expression profiles. The recurrence-free survival data includes recurrence-free survival status ("Recurred" or "Not recurred") and recurrence-free survival months.

### Renormalizing gene expression profiles

Among 2,509 samples in the METABRIC dataset and 817 samples in the TCGA dataset, we picked out 679 and 415 luminal-A breast cancer samples from each dataset based on the

PAM50 subtype data. We also selected 17,202 genes of which expression values are available in the all 679 METABRIC samples and 415 TCGA samples. Then, we chose the top 5,000 genes with the highest variability based on the median absolute deviations of each gene. Next, we renormalized the normalized gene expression profiles (median Z-scores) of the 679 METABRIC samples such that each $i$th gene expression value of the $j$th sample, $e^{j,META(i)}$, to be in the range between zero and one, and it could be calculated as:

$$X_1^{i,META(j)} = \frac{e^{i,META(j)} - \min(e^{i,META})}{\max(e^{i,META}) - \min(e^{i,META})} \tag{1}$$

where min $(e^{j,META(i)})$ and max $(e^{i,META(i)})$ is the minimum and maximum expression value of $i$th gene across the 679 METABRIC samples.

Similarly, we renormalized the normalized gene expression profiles of the 415 TCGA samples using the minimum and maximum expression value of each gene in the METABRIC dataset, and it could be calculated as:

$$X_1^{i,TCGA(j)} = \frac{e^{i,TCGA(j)} - \min(e^{i,META})}{\max(e^{i,META}) - \min(e^{i,META})} \tag{2}$$

We used python machine learning library Scikit-learn (version 0.23.2) to renormalize the gene expression profiles.

## Training deep autoencoders

We constructed deep autoencoders which are composed of five layers: input layer, three hidden layers, and output layer. As shown in Fig 1A, the renormalized gene expression profiles of each sample $X_1^j$ were used as the input features, and so the size of the input layer and the output layer was set as the number of genes (N = 5,000). Notably, we set the number of nodes same in all three hidden layers and trained the eight deep autoencoders with the different size of hidden layers (16, 32, 64, 128, 256, 512, 1024, and 2048) to see whether the size of hidden layers affect the performance of the deep autoencoders [19]. We used the ReLU function and the sigmoid function as an activation function in the hidden layers and the output layer, respectively. The latent features of $j$th sample generated in the $k$th layer of the deep autoencoders of which the hidden layer size is $dim$ could be calculated as:

$$X_{k,dim}^j = f_k(W_{k,dim}X_{k-1,dim}^j + b_{k,dim}), k \in \{3, 4, 5\} \tag{3}$$

In Eq (3), $f_k$ is the activation function in the $k$th layer. $W_{k,dim}$ and $b_{k,dim}$ are weight matrix and bias in the $k$th layer of the deep autoencoders of which the hidden layer size is $dim$. For example, $X_{3,64}^{META(j)}$ is the latent features of $j$th sample in the METABRIC dataset, which is generated in the second hidden layer of deep autoencoder of which the hidden layer size is 64.

We used the 679 METABRIC samples as a training set and the 415 TCGA samples as a validation set. We evaluated the performance of the deep autoencoders using mean squared error (MSE) which measures the differences between the renormalized gene expression profiles of the input layer ($X_1$) and the reconstructed gene expression profiles of the output layer ($X_5$), and the deep autoencoders were trained to minimize the MSE:

$$MSE = \frac{1}{NM} \sum_{j=1}^{M} \sum_{i=1}^{N} (X_5^{i,j} - X_1^{i,j})^2 \tag{4}$$

where N is the number of genes, and M is the number of samples. We used ADAM as an optimizer [46], and set the batch size and the epoch number as 16 and 100, respectively. All

procedures related to the construction and training of deep autoencoders were performed by python machine learning library Tensorflow (version 2.3.0).

## Dividing the samples into subgroups

As shown in Fig 1B, we generated the latent features of each 679 METABRIC sample in the second hidden layer of each four deep autoencoder ($dim\in$ 16, 32, 64, and 128). They were calculated as:

$$X_{3,dim}^{META(j)} = f_3\left(W_{3,dim}X_{2,dim}^{META(j)} + b_{3,dim}\right) \ where \ X_{2,dim}^{META(j)} = f_2\left(W_{2,dim}X_{1,dim}^{META(j)} + b_{2,dim}\right) \tag{5}$$

In consequence, each sample had the four latent features of the different dimensional size according to the size of hidden layers. Then, we divided the samples into several subgroups using each latent features of the different dimensional size as input features of K-Means clustering. We set the number of clusters as from two to five. Python machine learning library Scikit-learn (version 0.23.2) was used for the unsupervised learning.

## Comparing recurrence-free survival rate between subgroups

We performed Kaplan-Meier survival analysis [47] to compare prognosis between the subgroups using recurrence-free survival status and months data. The significance of prognostic difference was statistically estimated by log-rank test [48]. Specifically, we conducted both multivariate log-rank test and pairwise log-rank test to find the number of subgroups which shows the most distinct prognostic differences. In this step, we chose the size of hidden layers that show the most distinct prognostic differences between the subgroups for validation in the following step. The Kaplan-Meier survival analysis was implemented through python survival analysis library Lifelines (version 0.24.1).

## Validating the prognostic difference between BPS-LumA and WPS-LumA using an independent dataset

Using the renormalized gene expression profiles of 415 TCGA samples $X_1^{TCGA(j)}$, we generated the latent features of 415 TCGA samples in the second hidden layer of deep autoencoder of which the hidden layer size is 64. They could be calculated as:

$$X_{3,64}^{TCGA(j)} = f_3\left(W_{3,64}X_{2,64}^{TCGA(j)} + b_{3,64}\right) \ where \ X_{2,64}^{TCGA(j)} = f_2\left(W_{2,64}X_{1,64}^{TCGA(j)} + b_{2,64}\right) \tag{6}$$

Then, we assigned the each TCGA sample to the closest subgroup based on distance from the centroid of each subgroup in the latent space, which was identified using the samples in the METABRIC dataset. The Kaplan-Meier survival analysis was performed to compare prognosis between the samples belonging to each subgroup.

## Comparing stratifications with the previous studies

We compared our stratification with the two previous studies: Netanely's method [12] and Poudel's method [14]. We used the TCGA dataset for comparison because two previous studies provided their subgroup data of the TCGA samples. We obtained the subgroup data of TCGA samples from the supplemental information of original publications [12, 14]. We examined how significantly each previous method is consistent with our method by chi-squared test using python scientific computing library Scipy (version 1.4.1).

## Finding differentially expressed genes between BPS-LumA and WPS-LumA

We used the normalized gene expression profiles of METABRIC dataset and R package limma to find differentially expressed genes (DEGs) between the subgroups (p-value < 0.01) [20]. We performed the same procedures using the normalized gene expression profiles of the TCGA dataset. Then, we selected genes that are differentially expressed in both datasets among the top 5,000 genes with the highest variability that are used in training the deep autoencoders.

## Constructing weighted co-expression network of breast tissue

We constructed weighted co-expression network, which is able to represent normal breast tissue, and detected co-expressed modules using gene expression profiles obtained from GTEx (the Genotype-Tissue Expression) [49] and R package WGCNA (Weighted gene co-expression network analysis) [21, 22]. We downloaded gene read counts and TPMs (Transcripts Per Kilobase Million) profiles of 459 breast tissue obtained from non-diseased tissue sites. According to the median read count number per gene, we only considered the genes of which median read count number is larger than ten. We divided the 459 samples into training set (80%) and test set (20%). We used the training set to determine parameters of WGCNA to construct the weighted co-expression network and used the test set to test the significance of co-expressed module preservation. As a result, we constructed weighted co-expression network of breast tissue consisted of the 23 modules and 8,383 genes which are strongly preserved in the test set. In addition, because WGCNA provides the weights of every gene pair (fully connected), we only used top 10% edges with the highest weights and their nodes in each module.

## Performing gene-set enrichment analysis

We performed Enrichr [26] to find the biological functions associated with each co-expressed modules using gene-set enrichment analysis [50]. It was performed using the genes in each co-expressed module and gene-sets from Kyoto Encyclopedia of Genes and Genomes (KEGG) pathways [51], Wiki-Pathways [52], and Gene Onology [53] and we only considered the terms of which adjusted p-values are lower than 0.0001. Python gene set enrichment analysis library Gseapy (version 0.10.4) were used.

## Supporting information

**S1 Fig. The Kaplan-Meier survival analysis according to the dimensional size of latent features in the METABRIC dataset (the number of subgroups = 2).**
(DOCX)

**S2 Fig. The Kaplan-Meier survival analysis according to the number of clusters in the METABRIC dataset (64-dimensional latent features).**
(DOCX)

**S1 Table. The list of top 5,000 genes with the highest variability in the METABRIC dataset.**
(XLSX)

**S2 Table. The mean squared error (MSE) according to the size of hidden layers.**
(XLSX)

**S3 Table. The p-value of pairwise log-rank test according to the number of clusters (64-dimensional latent features).**
(XLSX)

**S4 Table. The subgroup data (BPS-LumA and WPS-LumA) of the 679 METABRIC luminal-A breast cancer samples.**
(XLSX)

**S5 Table. The subgroup data (BPS-LumA and WPS-LumA) of the 415 TCGA luminal-A breast cancer samples.**
(XLSX)

**S6 Table. The list of differentially expressed genes between the BPS-LumA and WPS-LumA in the METBRIC dataset.**
(XLSX)

**S7 Table. The list of differentially expressed genes between the BPS-LumA and WPS-LumA in the TCGA dataset.**
(XLSX)

**S8 Table. The results of co-expressed module preservation tests.**
(XLSX)

**S9 Table. The list of genes belonged to each co-expressed module.**
(XLSX)

**S10 Table. The proportion of genes overlapping with DEGs in each co-expressed module.**
(XLSX)

**S11 Table. The results of gene-set enrichment analysis of the co-expressed modules.**
(XLSX)

**S12 Table. The comparison with the previous luminal-A breast cancer stratification methods.**
(XLSX)

## Author Contributions

**Conceptualization:** Seunghyun Wang, Doheon Lee.

**Data curation:** Seunghyun Wang.

**Formal analysis:** Seunghyun Wang, Doheon Lee.

**Funding acquisition:** Doheon Lee.

**Investigation:** Seunghyun Wang, Doheon Lee.

**Methodology:** Seunghyun Wang, Doheon Lee.

**Project administration:** Seunghyun Wang, Doheon Lee.

**Resources:** Seunghyun Wang, Doheon Lee.

**Software:** Seunghyun Wang.

**Supervision:** Doheon Lee.

**Validation:** Seunghyun Wang.

**Visualization:** Seunghyun Wang.

**Writing – original draft:** Seunghyun Wang, Doheon Lee.

**Writing – review & editing:** Seunghyun Wang, Doheon Lee.

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
