## [Decision Letter · Decision Letter 0]

28 Feb 2023

Dear Dr. Lee,

Thank you very much for submitting your manuscript "Identifying prognostic subgroups of luminal-A breast cancer using deep autoencoders and gene expressions" for consideration at PLOS Computational Biology.

As with all papers reviewed by the journal, your manuscript was reviewed by members of the editorial board and by several independent reviewers. In light of the reviews (below this email), we would like to invite the resubmission of a significantly-revised version that takes into account the reviewers' comments.

Both reviewers have serious concerns about the methods and results described in the paper.

We cannot make any decision about publication until we have seen the revised manuscript and your response to the reviewers' comments. Your revised manuscript is also likely to be sent to reviewers for further evaluation.

Sincerely,

Mark Alber, Ph.D.

Section Editor

PLOS Computational Biology

Mark Alber

Section Editor

PLOS Computational Biology

Reviewer's Responses to Questions

**Comments to the Authors:**

Reviewer #1: The authors suggest an autoencoder model that stratifies subgroups of breast cancers that displays different prognostic outcomes. They built an autoencoder model that takes gene expression data of breast cancer samples for feature extraction, then performed k-means clustering to identify subgroups, which indeed show different survival profiles in independent TCGA data as well as the METABRIC training data. They also compared the stratification performance against conventional models that directly use gene expression profiles as input with a varying number of genes. Further, they sought to explain the prognostic outcomes by overlaying differentially expressed genes between groups onto the co-expression network constructed from independent gene expression data from healthy samples. Meanwhile, it is a novel approach to delineating complicated biological samples, this work requires technical clarification and analysis.

Autoencoder is indeed widely used for feature extraction and is proven to be effective in many studies. However, the actual application here is with some concern. First, the authors used all detected genes as input. Authors argue that it minimizes manual filtering and intervention. But obviously, there should be many genes that display minimal variance, hence are uninformative. These genes will be more of noise rather than useful features. This issue is related to the way of re-normalization by min/max scaling. Even when a gene displays very minimal noisy variation, those variances will be amplified via min/max scaling and ultimately will have an equal contribution to the model as much as highly variable and informative genes, which will obscure successful learning of the model. Also related to this in Fig4, the p-value vividly decreases as the number of input genes decreases, which suggests a preliminary gene filtering can be beneficial. Therefore, I’d like to recommend trying a combined approach of preliminary gene filtering and autoencoder, i.e. low variance gene filtering followed by autoencoder training. Here, I’d recommend selecting highly variable genes after variance stabilization to avoid the preference of highly expressed genes. If successful, it will help us to achieve a more compact model.

On a second note, it is somewhat surprising to see that an autoencoder model trained using array data successfully classifies RNA-seq data. Is it because of the min-max normalization and highly correlated array vs RNA-seq data? Please discuss.

Also, the authors build a WGCNA model using seemingly all detected genes. Given the nature of WGCNA relying on correlation coefficient, it conveys a similar risk when involving low variance genes. How would the author justify this? What if similar preliminary gene filtering is applied?

In Fig 6

- Panel B and C are not quite informative. Rather I’d recommend putting 1) a bar plot module size, and 2) % of genes in each module overlapping with DEG

- Panels seem to be mislabeled in the legend for B and C. A appears twice, B once, and no C.

In line 413, the authors described that the hidden layer size is 2048. However, they tried all different numbers of hidden layers in the supplementary figures. Please correct if necessary.

Reviewer #2: In this paper, the authors proposed a data-driven method for identifying prognostic markers that can differentiate between different groups of breast cancer patients with different prognosis.

The main idea is to first train auto-encoders that can extract low-dimensional latent features that encode the gene expression data from breast cancer patients and then can be used to faithfully reconstruct the gene expression data.

The extracted features are then used to cluster breast cancer patients using k-means algorithm, and the authors demonstrate that the identified subgroups show very different prognosis.

While the proposed method and the presented results are reasonable, there are several major concerns in the manuscript that need to be addressed.

These concerns are outlined below.

1. While the use of autoencoder to extract potential latent features for stratifying breast cancer patients is reasonable, the proposed method lacks novelty.

The authors mainly train a standard autoencoder with different latent dimensions using breast cancer gene expression data, which is technically not novel.

Except for varying the dimension size, no other architecture/hyperparameter optimization is performed to customize the model for the task and enhance the stratification results.

2. The authors show that using the latent dimension of 2048 led to the best (most significant results).

But considering that this latent dimension size was the largest among the test dimension sizes, and as this dimension size is comparable to the dimension of the original gene space, it is highly likely that the model is overfitted to the data (especially, since the sample size of the gene expression data was relatively small compared to the latent dimension).

3. The latent features do not seem to have any biological significance. For example, while there has been recent work on constraining the latent space based on pathway info - in which case the latent features may potentially reflect underlying pathway activities, in this current work it is difficult to see whether the 2,048 latent features meaningfully reflect the underlying biological processes/pathways that may be associated with different prognosis of subgroups.

Although the authors perform some analysis by analyzing the differentially expressed genes between the subgroups that are identified by the proposed method, such analysis beats the purpose of using an auto-encoder for latent feature extraction in the first place.

For example, we could have simply started with DEG analysis without using autoencoders to detect gene markers that can be used to stratify subgroups and then analysis the identified DEGs to understand what might be leading to this difference.

Why should one first use autoencoders to extract latent features (that cannot be easily interpreted), and then resort to DEG analysis for interpretation?

4. Figure 2 and Figure 3 show PCA plots for the latent feature vectors, but how does it compare to using PCA directly to gene expression data?

Why not use PCA to extract important principal components that reflect the main gene expression patterns, use these principal components for clustering patients and stratifying them?

Does the use of autoencoder significantly improve the results?

Such comparison would be critical in demonstrating the effectiveness of the proposed scheme.

5. Furthermore, there have been pathway based methods and network analysis based methods for extracting gene modules that may be used as diagnostic and/or prognostic markers.

How does the use of autoencoder-based data driven scheme for extracting latent features compare with more traditional methods that define/predict modular markers based on pathways or network info?

Is there a clear evidence that even without using such additional information (i.e., pathway info or network info), the autoencoder can extract better prognostic signatures thare are predictive of patients' survival?

6. The authors claim that "our method and previous methods can be complementary to each other for a more precise stratification for luminal-A breast cancer". However, this conclusion is drawn simply based on the fact the difference/discrepancies between the different methods.

To justify such statement, the authors should actually show "how" they could be combined to improve the stratification results.

7. (minor comment) The literary presentation of the paper should be improved, as the current manuscript includes a large number of grammatical and typographical errors.

8. (minor comment) in line 198, the author mention that "the two datasets used the different sequencing platform", but note that microarray is not a "sequencing platform".

**Have the authors made all data and (if applicable) computational code underlying the findings in their manuscript fully available?**

Reviewer #1: Yes

Reviewer #2: Yes

PLOS authors have the option to publish the peer review history of their article (what does this mean?). If published, this will include your full peer review and any attached files.

Reviewer #1: No

Reviewer #2: No
---

## [Decision Letter · Decision Letter 1]

30 Apr 2023

Dear Dr. Lee,

Thank you very much for submitting your manuscript "Identifying prognostic subgroups of luminal-A breast cancer using deep autoencoders and gene expressions" for consideration at PLOS Computational Biology. As with all papers reviewed by the journal, your manuscript was reviewed by members of the editorial board and by several independent reviewers. The reviewers appreciated the attention to an important topic. Based on the reviews, we are likely to accept this manuscript for publication, providing that you modify the manuscript according to the review recommendations.

Sincerely,

Mark Alber, Ph.D.

Section Editor

PLOS Computational Biology

Mark Alber

Section Editor

PLOS Computational Biology

Reviewer's Responses to Questions

**Comments to the Authors:**

Reviewer #1: Authors addressed all the concerns.

One minor comment:

Authors changed the presentation of the dimension reduction in the main figures. What was the basis of this change

Reviewer #2: I would like to thank the authors for their careful revision.

Most of my concerns regarding the initial version of this manuscript have been sufficiently addressed in the revised version, and I believe the main advantages of the proposed approach are conveyed in a clearer and a more convincing manner.

For example, the updated analysis based on a lower-dimensional latent representation addresses the concern regarding overfitting, and the updated analysis results and discussions are therefore more convincing.

Direct comparison between the proposed autoencoder-based scheme and the traditional PCA based dimensionality reduction is also very helpful, and it clearly demonstrates how using deep network models for dimensionality reduction can identify useful latent features that can more accurately cluster and stratify different patient groups with different prognosis, which may be difficult based on traditional schemes like the PCA.

Regarding my previous comment concerning the comparison against existing methods that used prior knowledge (e.g., pathways or PPI networks), while I understand the authors position I nevertheless think that it would be meaningful to include at least some discussion comparing the pros and cons of the respective approaches.

Finally, the authors have revised a number of places in the manuscript to convey the novel contributions made in this work more clearly, these are spread over the paper and it may be a good idea to summarize the main novel contributions more clearly at the outset (e.g., in the last paragraph of the introduction section).

For example, the authors may want to clearly mention that while they are not proposing a novel methodology, the application of an autoencoder to luminal-A breast cancer to extract biologically meaningful low-dimensional latent features is novel and it may detect features not possible using other traditional techniques.

**Have the authors made all data and (if applicable) computational code underlying the findings in their manuscript fully available?**

Reviewer #1: Yes

Reviewer #2: Yes

PLOS authors have the option to publish the peer review history of their article (what does this mean?). If published, this will include your full peer review and any attached files.

Reviewer #1: No

Reviewer #2: No

Figure Files:

Data Requirements:

Reproducibility:

References:

---

## [Decision Letter · Decision Letter 2]

18 May 2023

Dear Dr. Lee,

We are pleased to inform you that your manuscript 'Identifying prognostic subgroups of luminal-A breast cancer using deep autoencoders and gene expressions' has been provisionally accepted for publication in PLOS Computational Biology.

Best regards,

Mark Alber, Ph.D.

Section Editor

PLOS Computational Biology

Mark Alber

Section Editor

PLOS Computational Biology

Reviewer's Responses to Questions

**Comments to the Authors:**

Reviewer #1: Acceptable for publication.

Reviewer #2: Thank you for the revision.

The previous revision was fairly comprehensive and it already addressed all major concerns I had regarding the original manuscript.

The remaining concerns were mostly minor, and the current revision has addressed them in a satisfactory manner.

I do not have any further suggestions.

**Have the authors made all data and (if applicable) computational code underlying the findings in their manuscript fully available?**

Reviewer #1: Yes

Reviewer #2: None

PLOS authors have the option to publish the peer review history of their article (what does this mean?). If published, this will include your full peer review and any attached files.

Reviewer #1: No

Reviewer #2: No

---

## [Editor Report · Acceptance letter]

25 May 2023

PCOMPBIOL-D-23-00002R2 

Identifying prognostic subgroups of luminal-A breast cancer using deep autoencoders and gene expressions

Dear Dr Lee,

I am pleased to inform you that your manuscript has been formally accepted for publication in PLOS Computational Biology. Your manuscript is now with our production department and you will be notified of the publication date in due course.

With kind regards,

Anita Estes
